# Learning Fuel-Optimal Trajectories for Space Applications via Pontryagin Neural Networks

Andrea D'Ambrosio [1] and Roberto Furfaro [1,2,*]

1   Systems & Industrial Engineering, University of Arizona, Tucson, AZ 85721, USA; dambrosio@email.arizona.edu
2   Aerospace & Mechanical Engineering, University of Arizona, Tucson, AZ 85721, USA
*   Correspondence: robertof@arizona.edu; Tel.: +1-520-621-2525

**Abstract:** This paper demonstrates the utilization of Pontryagin Neural Networks (PoNNs) to acquire control strategies for achieving fuel-optimal trajectories. PoNNs, a subtype of Physics-Informed Neural Networks (PINNs), are tailored for solving optimal control problems through indirect methods. Specifically, PoNNs learn to solve the Two-Point Boundary Value Problem derived from the application of the Pontryagin Minimum Principle to the problem's Hamiltonian. Within PoNNs, the Extreme Theory of Functional Connections (X-TFC) is leveraged to approximate states and costates using constrained expressions (CEs). These CEs comprise a free function, modeled by a shallow neural network trained via Extreme Learning Machine, and a functional component that consistently satisfies boundary conditions analytically. Addressing discontinuous control, a smoothing technique is employed, substituting the sign function with a hyperbolic tangent function and implementing a continuation procedure on the smoothing parameter. The proposed methodology is applied to scenarios involving fuel-optimal Earth−Mars interplanetary transfers and Mars landing trajectories. Remarkably, PoNNs exhibit convergence to solutions even with randomly initialized parameters, determining the number and timing of control switches without prior information. Additionally, an analytical approximation of the solution allows for optimal control computation at unencountered points during training. Comparative analysis reveals the efficacy of the proposed approach, which rivals state-of-the-art methods such as the shooting technique and the adaptive Gaussian quadrature collocation method.

**Keywords:** fuel optimal trajectories; machine learning; Pontryagin Neural Networks; Physics-Informed Neural Networks; Extreme Theory of Functional Connections; optimal control

## 1. Introduction

Addressing optimal control problems (OCPs) stands as a critical endeavor in crafting precise and efficient maneuvers for space missions. Traditionally, OCPs are approached through either direct or indirect methods. The direct method entails discretizing the problem to transform it into a Non-Linear Programming (NLP) problem, which is subsequently solvable via established optimization techniques like the trust region method, Nelder–Mead method, or interior point method [1]. However, a significant concern with direct methods arises from the fact that general NLP problems are deemed NP-hard, implying indeterminate polynomial time complexity. This characteristic entails that the computational effort required to attain the optimal solution lacks a predefined bound, with no assurance of optimality or convergence, thereby casting doubt on the reliability of these approaches. Conversely, the indirect method harnesses the calculus of variation and the Pontryagin Minimum Principle (PMP) [2]. Consequently, first-order necessary conditions for optimality are derived from the problem's Hamiltonian in terms of states and costates. This leads to the formulation of Ordinary Differential Equations (ODEs), constituting the Two-Point Boundary Value Problem (TPBVP), typically tackled through single and multiple shooting

methods [3,4], orthogonal collocation [5], or pseudo-spectral methods [6]. However, a notable drawback of the indirect optimization lies in its narrow convergence window, which is heavily reliant on the initial guess of unknown initial costates. Furthermore, costates often lack direct physical interpretation, exacerbating the challenge of estimating an initial guess. Consequently, despite the theoretical guarantees of optimality, obtaining optimal control via the indirect method can prove arduous.

Among the OCPs employed for space exploration missions, minimum time trajectories and fuel optimal trajectories, which are also considered in this paper, play a key role especially during the orbit transfer phases and the proximity maneuvers around a target body. Applications of such optimal trajectories already studied in the literature include cislunar trajectories [7–11], low−thrust orbit transfer around the Earth [12], interplanetary trajectories [13], and solar sail optimal trajectories [14]. Specifically, the primary problem centers on reducing propellant consumption (or maximizing final mass), whereas the secondary objective is to minimize the time of flight. The common aspect between those two types of problem is that a discontinuous control is usually involved when traditional thrusters are employed. This occurs because the control input, often represented by the throttle factor (or engine thrust ratio), appears linearly in the Hamiltonian of the problem, thus leading to a bang−off−bang or bang−bang type of control. In particular, two major difficulties arise when dealing with fuel optimal problems tackled via an indirect method. First, an estimate for initializing the costates is necessary, and at times, it can be challenging to provide such an estimate due to the lack of direct physical interpretation for the costates. Secondly, the number of switches of the control and their temporal location are usually unknown. To deal with that, a very accurate and extensive study is presented by Taheri and Junkins [15] with the goal of generating minimum-fuel switching surfaces and computing the solution of N-impulse fuel optimal interplanetary rendezvous and Earth orbit transfers. Moreover, the link between impulsive and continuous-thrust trajectories is also proved via optimal switching surfaces. Due to the strong importance of fuel-optimal (and minimum time) trajectories, many other works in the literature have been dedicated to the study of those trajectories and how to mitigate the difficulties arising when dealing with discontinuous control. In particular, three main techniques can be highlighted (eventually used in combination): the homotopic continuation procedure, the convexification technique, and the smoothing function technique.

The homotopic continuation procedure is widely employed and effective in solving fuel or time-optimal problems [10–14]. It consists of linking the original (difficult) OCP with easier problems to solve. This technique is based on a homotopic continuation parameter whose value is usually equal to one to represent the easier OCP. Thus, this problem is first solved, and then more difficult OCPs are solved step by step by slowly decreasing the continuation parameter to low values (close to zero). This continuation procedure allows for obtaining an accurate solution of the original OCP. Within homotopy continuation, three perturbing functions are introduced by Bertrand and Epenoy [16]: the quadratic, logarithmic and extended logarithmic functions. Considering different types of perturbing functions actually leads to having different formulations of the optimal throttle input, while the optimal thrust unit direction always remains the same. Regarding the quadratic perturbing function, it serves as a bridge connecting an energy-optimal problem to a fuel-optimal problem and has been utilized in numerous studies. To cite some examples, it was employed in [17] to derive low−thrust fuel−optimal trajectories, in [18] to calculate fuel−optimal low−thrust Earth−orbit transfers, accounting for shadow eclipses, and in [19] to study fuel optimal soft landing trajectories on asteroids. As an example of the logarithmic perturbing function, it is exploited by Izzo and Öztürk [20] to obtain the fuel−optimal trajectories required to build a dataset and train a Deep Neural Network (DNN) in a supervised fashion. The resulting model appears to hold promise for the potential real−time onboard implementation of an optimal guidance and control system for a spacecraft. Even in the case of a homotopic continuation procedure, an initial guess of the initial costates is still required. A methodology to approximate the initial

costates for a fuel−optimal descent trajectory on asteroids is proposed in Ref. [21], where a two−impulse descent trajectory computed via an irregular gravitational Lambert solver is used for the costates initialization, thus removing the issue and showing feasible results in low computational times. Continuation procedures are also employed in [22,23]. The initial paper achieves fuel−optimal orbital transfers by employing Lawden's primer vector theory and implementing a continuation procedure on the thrust amplitude. Additionally, the proposed strategy enables the development of an automated algorithm, offering the benefit of not necessitating any initial guess for the costate variables. In the subsequent study, a novel homotopy continuation technique is introduced, connecting the original fuel−optimal low−thrust trajectory with the time−optimal problem. Moreover, the dynamical model introduces new variables to reduce the number of unknown initial costates, allowing the mass costate to be expressed analytically in logarithmic form.

Another important methodology to rapidly and accurately solve OCPs is the convexification technique, which allows for transforming nonconvex problems into convex problems. In fact, convex problems are easier to solve, and theoretical guarantees about the solutions convergence and the computational efficiency are generally available. Convexification has been exploited widely within the aerospace community for fuel−optimal problems, involving (but not limited to) the landing on Mars [24] and asteroids [25], transfer trajectories between periodic orbits in the cislunar space [26], cooperative rendezvous [27], and interplanetary low−thrust trajectories using a subsequent optimization process [28]. Finally, fuel−optimal and minimum time trajectories are linked and computed via convex optimization in [29], where the authors first solve fuel−optimal trajectories in order to compute accurate minimum time trajectories. However, for further details about convex optimization for aerospace applications, the reader can refer to Ref. [30].

Finally, smoothing techniques are based on the approximation of the sign function involved in the discontinuous control with smooth functions. Even in this case, there is the presence of a smoothing parameter which is slowly decreased with a continuation procedure to accurately obtain the discontinuous control. Many smoothing techniques have been proposed in the literature. As an example, a trigonometric−based regularization is employed in [31] to study fuel−optimal trajectories including also path constraints. The same kind of problem is faced in Refs. [32–34], where a hyperbolic tangent smoothing function is employed. This last smoothing function is actually the smoothing technique exploited in this paper to approximate the discontinuous control.

This work delves into fuel−optimal trajectories with fixed time of flight, tackled through the combination of indirect methods and a machine learning approach known as Pontryagin Neural Networks (PoNNs), which is a specialized framework within Physics-Informed Neural Networks (PINNs). As defined in [35], PoNNs are a subset of PINNs specifically trained to learn optimal control actions conforming to the Pontryagin Minimum Principle (PMP). By leveraging PoNNs, solutions to the Two−Point Boundary Value Problems (TPBVPs) associated with fuel−optimal scenarios are learned in terms of states and costates. Notably, the PINN framework utilized in PoNNs is the Extreme Theory of Functional Connections (X−TFC), which combines the functional interpolation technique known as the Theory of Functional Connections (TFC), pioneered by Mortari [36], and the Extreme Learning Machine (ELM) [37]. According to TFC, latent solutions are represented by constrained expressions (CEs), comprising a free function and a functional component that consistently satisfies boundary conditions analytically. The analytical fulfillment of these boundary constraints offers a significant advantage in solving TPBVPs.

Within X−TFC, the free function is represented by a shallow neural network (NN) trained via Extreme Learning Machine (ELM). Notably, ELM is a training algorithm wherein input weights and biases are randomly sampled from continuous distributions and remain untuned throughout training. Consequently, the only parameters adjusted during training are the output weights. Typically, least−square (LS) methods are employed within ELM for training, with proofs of convergence provided in [37]. In this work, Chebyshev Neural Networks (ChNN) are utilized as the free−function [38]. X−TFC emerges as a versatile

tool applicable to various domains, ranging from data−driven parameter discovery of Ordinary Differential Equations (ODEs) [39] to solving zero−finding problems by identifying promising homotopy paths [40]. Furthermore, owing to their efficacy in solving Two−Point Boundary Value Problems (TPBVPs), both frameworks have been employed in solving optimal control problems (OCPs) within aerospace applications. For instance, they have been utilized in energy−optimal landing problems on small and large planetary bodies [41,42], energy−optimal circumnavigation trajectories around asteroids with collision avoidance [43], energy−optimal relative motion problems [44], optimal planar orbit transfers [45], and intercept problems [35]. However, in a previous work addressing fuel−optimal landing on large planetary bodies with constant gravity through TFC [46], the number of control switches was known in advance, allowing for the problem to be simplified by explicitly segmenting the time domain into three parts. Conversely, in the present study, the number of switches for discontinuous control is not assumed to be known a priori. To the best of the authors' knowledge, this study marks the first instance where PoNNs are employed to learn solutions of OCPs featuring discontinuous control in aerospace applications. Specifically, two distinct fuel−optimal problems are tackled to evaluate the proposed approach: a low−thrust interplanetary transfer from Earth to Mars orbit and a landing trajectory on Mars. For both scenarios, the obtained solutions are compared with other state−of−the−art methods, such as the shooting method and adaptive Gaussian quadrature collocation method.

Finally, the main contributions of this paper include the following: (1) the extension of the PoNN framework to generate solutions of OCPs with discontinuous control thanks to the combination with the smoothing hyperbolic tangent and the continuation procedure; (2) the proposed framework autonomously detects the number of switches in the control as well as their temporal location without any a priori knowledge; (3) the solution convergence is obtained with random initial guesses of the output weights of the PoNN; (4) the utilization of the X−TFC constrained expressions (CEs) facilitates the availability of an analytical approximation for the optimal trajectory and control, obviating the need for interpolation to compute solutions at points not encountered during training and mitigating potential accuracy degradation.

This paper is organized as follows. Section 1 is dedicated to a brief recall of the indirect method and the presentation of the proposed strategy to solve OCPs via PoNNs. Afterwards, fuel−optimal problems are formulated for both the interplanetary transfer and the landing trajectory, and the latent solutions approximation via CEs is provided. Sections 3 and 4 report the obtained results and related discussions, respectively. Section 6 provides concluding remarks.

## 2. Pontryagin Neural Networks

This section offers a concise overview of tackling optimal control problems (OCPs) using the indirect method. Initially, we outline the derivation of the Two−Point Boundary Value Problem (TPBVP) from the application of the Pontryagin Minimum Principle (PMP) and the calculus of variations. Subsequently, we delve into the design and training process of Pontryagin Neural Networks (PoNNs), focusing on their ability to learn the state−costate pair, which constitutes the solution to the TPBVP.

### 2.1. Optimal Control Problems via Indirect Method

Optimal control problems (OCPs) typically entail a system of differential equations governing the evolution of state and control variables while also adhering to an optimal criterion expressed through the minimization or maximization of a cost function. Broadly, this cost function is contingent upon both state and control variables, which are represented as follows:

$$\mathcal{J} = \Phi(\boldsymbol{x}(t_0), t_0, \boldsymbol{x}(t_f), t_f) + \int_{t_0}^{t_f} \mathscr{L}(\boldsymbol{x}(t), \boldsymbol{u}(t), t)\, dt \tag{1}$$

subject to the dynamic constraints, expressed as

$$\dot{\boldsymbol{x}} = \boldsymbol{f}(\boldsymbol{x}(t), \boldsymbol{u}(t), t) \tag{2}$$

and the boundary conditions

$$\boldsymbol{\Phi}(\boldsymbol{x}(t_0), t_0) = \boldsymbol{\Phi}_0 \tag{3}$$
$$\boldsymbol{\Phi}(\boldsymbol{x}(t_f), t_f) = \boldsymbol{\Phi}_f \tag{4}$$

In the previous equations, $\boldsymbol{x}(t)$ represents the state vector, $\boldsymbol{u}(t)$ signifies the control vector, and $t$ denotes the independent variable, which is typically time. The parameters $t_0$ and $t_f$ correspond to the initial and final time instants, respectively. Within Equation (1), $\Phi$ denotes the end−point cost, often referred to as the Meyer cost, while $\mathscr{L}$ represents the running cost, alternatively known as the Lagrangian cost [1]. As mentioned earlier, optimal control problems can be addressed using either direct or indirect methods. The methodology proposed in this study relies on the indirect approach. Consequently, the optimal control problems are tackled by employing the Pontryagin Maximum (or Minimum) Principle (PMP) [47]. Utilizing the PMP necessitates the formulation of the Hamiltonian, which entails

$$H = \mathscr{L} + \boldsymbol{\lambda}^{\mathsf{T}} \boldsymbol{f} \tag{5}$$

where $\boldsymbol{\lambda}$ represent the costate (or adjoint variables). Following the first−order optimality conditions of the Pontryagin Maximum (or Minimum) Principle (PMP), the optimal control can be obtained by differentiating the Hamiltonian with respect to the control vector and equating it to zero:

$$\frac{\partial H}{\partial \boldsymbol{u}} = 0 \tag{6}$$

Furthermore, by applying the first−order necessary conditions for the state and costate variables, we derive the following system of ODEs:

$$\dot{\boldsymbol{x}} = \frac{\partial H}{\partial \boldsymbol{\lambda}} \tag{7}$$

$$\dot{\boldsymbol{\lambda}} = -\frac{\partial H}{\partial \boldsymbol{x}} \tag{8}$$

Lastly, transversality conditions on the costates and the Hamiltonian, if applicable (e.g., when the corresponding state variable is unconstrained), must be enforced. To provide comprehensive coverage, the potential transversality conditions (excluding any constraints) are listed below:

$$\boldsymbol{\lambda}(t_0) = -\frac{\partial \mathcal{J}}{\partial \boldsymbol{x}_0} \tag{9}$$

$$H(t_0) = \frac{\partial \mathcal{J}}{\partial t_0} \tag{10}$$

$$\boldsymbol{\lambda}(t_f) = \frac{\partial \mathcal{J}}{\partial \boldsymbol{x}_f} \tag{11}$$

$$H(t_f) = -\frac{\partial \mathcal{J}}{\partial t_f} \tag{12}$$

Equations (6)−(8), coupled with the transversality conditions on the Hamiltonian, constitute a BVP whose solutions will be acquired using PoNNs. It is worth noting that with the proposed approach, the transversality conditions for the costate are inherently satisfied a priori, as elucidated later. Therefore, they do not explicitly feature in the BVP.

*2.2. X−TFC for TPBVPs*

In the PoNNs framework, the X−TFC framework is utilized to tackle a generic TPBVP in the time domain. The implicit form of the vector differential equation for a generic TPBVP in the time domain is expressed as follows:

$$F_i\left(t, y_j(t), \dot{y}_j(t), \ddot{y}_j(t)\right) = 0 \quad \text{subject to:} \begin{cases} y_j(t_0) = y_{0_j} \\ y_j(t_f) = y_{f_j} \\ \dot{y}_j(t_0) = \dot{y}_{j_0} \\ \dot{y}_j(t_f) = \dot{y}_{f_j} \end{cases} \tag{13}$$

In the previous equation, $i$ denotes the number of differential equations constituting the ODEs system, while $j$ represents the number of unknown functions $y_j(t)$, which serve as the solutions of the system. The independent variable is time $t$, belonging to the interval $[t_0, t_f]$. The initial phase of the X−TFC method involves deriving constrained expressions and their derivatives, akin to the process developed in the original TFC [36],

$$y_j^{(\ell)}(t) = g_j^{(\ell)}(t) + \sum_{k=1}^{n_j} \eta_{k_j} s_k^{(\ell)}(t) \tag{14}$$

In the given expression, the superscript $\ell$ denotes the $\ell$th derivative with respect to the independent variable. $n_j$ represents the number of constraints for the $j$th unknown function and/or its derivatives, $\eta_{k_j}$ denotes coefficients, and $g_j(t)$ signifies the free function. As outlined in [36], the functions $s_k(t)$, termed support functions, can be chosen as follows:

$$s_k(t) = t^{k-1} \tag{15}$$

After defining the constrained expression, enforcing the constraints on $y_j(t)$ and/or its derivatives at the boundary time instances (e.g., $t_0$ and $t_f$) within the constrained expression yields a system of linear algebraic equations. This system is subsequently solved to determine the coefficients $\eta_{k_j}$.

Once the $\eta_{k_j}$ coefficients are computed, the boundary constraints from Equation (13) are analytically embedded into the constrained expression. Subsequently, inserting the constrained expressions into the $F_i$ differential equations transforms them into a revised set of equations denoted as $\tilde{F}_i$. This revised set of equations solely depends on the independent variable $t$, the free function $g_j(t)$, and their derivatives. Specifically,

$$\tilde{F}_i\left(t, g_j(t), \dot{g}_j(t), \ddot{g}_j(t)\right) = 0 \tag{16}$$

The original constrained vector differential equation undergoes a transformation into an unconstrained vector differential equation. This occurs because the boundary conditions are embedded within it through the derived $\eta_{k_j}$ values. To address Equation (16), X−TFC employs a single−layer neural network (NN) as the free function, denoted as $g_j(t)$, which is trained using the Extreme Learning Machine (ELM) algorithm [37]. That is,

$$g_j(z) = \sum_{q=1}^{L} \beta_{j,q} \sigma_{j,q}(w_q z + b_q) = \begin{bmatrix} \sigma_{j,1} \\ \vdots \\ \sigma_{j,L} \end{bmatrix}^T \boldsymbol{\beta}_j = \boldsymbol{\sigma}_j^{\mathrm{T}}(z)\boldsymbol{\beta}_j \tag{17}$$

Here, $L$ represents the number of hidden neurons. $w_q \in \mathbb{R}$ denotes the input weight connecting the $q$th hidden neuron to the input nodes, while $\beta_{j,q} \in \mathbb{R}$, with $q = 1, \ldots, L$, represents the output weight connecting the $q$th hidden neuron to the output node. Additionally, $b_q$ stands for the bias of the $q$th hidden neuron. The function $\sigma_{j,q}(\cdot)$ signifies the activation functions chosen for the free function $g_j(z)$ with the same activation function

typically selected for all neurons and free functions. The reader should note that while the symbol $\sigma_{j,q}(\cdot)$ represents only the activation function, the bold symbol $\boldsymbol{\sigma}_j$ defines the entire hidden layer matrix.

Equation (17) delineates the crucial disparity between X−TFC and the standard TFC, underscoring why the X−TFC framework is classified within the family of Physics-Informed Neural Networks (PINNs). By employing a neural network (NN) as a free function, instead of orthogonal polynomials as in the standard TFC, the X−TFC approach achieves two significant advantages: (1) it substantially diminishes the curse of dimensionality in ODE and PDE problems compared to the standard TFC; (2) it can be categorized as a PINN method, aligning with the broader paradigm of utilizing neural networks to incorporate physics−based constraints into machine learning models.

Given that we employ the ELM algorithm to train the neural network (NN) [37], the only unknowns to compute are the output weights $\boldsymbol{\beta}_j = \begin{bmatrix} \beta_{j,1}, \ldots, \beta_{j,L} \end{bmatrix}^{\mathrm{T}}$. The attentive reader may observe the utilization of a different independent variable, $z$, rather than the original time variable. This deviation arises because the domains of the activation functions and the problem typically do not align. Consequently, we must map the domain $t$ into the domain $z$, and vice versa, to ensure compatibility:

$$z = z_0 + c(t - t_0) \quad \longleftrightarrow \quad t = t_0 + \frac{1}{c}(z - z_0) \tag{18}$$

where $c$ is a mapping coefficient, that is,

$$c = b^2 = \frac{z_f - z_0}{t_f - t_0} \tag{19}$$

Given that $c$ is always a positive number, it is advantageous to express it as $c = b^2$. Due to the mapping, all subsequent derivatives of $g_j(t)$ are defined as follows:

$$\frac{\mathrm{d}^n g_j}{\mathrm{d}t^n} = \boldsymbol{\beta}_j^{\mathrm{T}} \frac{\mathrm{d}^n \sigma_j(z)}{\mathrm{d}z^n} \left( \frac{\mathrm{d}z}{\mathrm{d}t} \right)^n = \boldsymbol{\beta}^{\mathrm{T}} \frac{\mathrm{d}^n \sigma_j(z)}{\mathrm{d}z^n} (b^2)^n \tag{20}$$

It is noteworthy that for optimal control problems where the final time is free, the mapping coefficient becomes an unknown quantity that must be determined alongside all the $\boldsymbol{\beta}_j$ output weights. The transformation of the free function and its derivatives from the $t$ domain to the $z$ domain can be summarized as follows:

$$\begin{cases} g_j(t) = \boldsymbol{\sigma}_j^{\mathrm{T}}(z)\boldsymbol{\beta}_j \\ \dot{g}_j(t) = b^2 \, \boldsymbol{\sigma}'^{\mathrm{T}}_j(z)\boldsymbol{\beta}_j \\ \ddot{g}_j(t) = b^4 \, \boldsymbol{\sigma}''^{\mathrm{T}}_j(z)\boldsymbol{\beta}_j \end{cases} \tag{21}$$

where $\sigma'_j(z)$ is the abbreviation for $\frac{\mathrm{d}\sigma_j(z)}{\mathrm{d}z}$. Equation (16) in the $z$ domain then becomes

$$\tilde{F}_i(z, \boldsymbol{\beta}_j) = 0 \tag{22}$$

To numerically address this TPBVP, we need to partition the $z$ domain into $n$ points. In this study, we discretize $z$ using evenly spaced points, although alternative quadrature

schemes can also be utilized. Subsequently, the unconstrained set of differential equations in Equation (22) can be represented as loss functions evaluated at each discretization point:

$$
\mathcal{L}_i(\boldsymbol{\beta}_j) = \left\{ \begin{array}{c} \tilde{F}_i(z_0, \boldsymbol{\beta}_j) \\ \vdots \\ \tilde{F}_i(z_d, \boldsymbol{\beta}_j) \\ \vdots \\ \tilde{F}_i(z_n, \boldsymbol{\beta}_j) \end{array} \right\}
\tag{23}
$$

By combining the differential equation for each dimension, an augmented loss function is formulated as follows:

$$
\mathbb{L} = \left\{ \mathcal{L}_1^{\mathrm{T}}, \quad ..., \quad \mathcal{L}_i^{\mathrm{T}}, \quad ..., \quad \mathcal{L}_{\mathcal{N}_{\mathrm{eq}}}^{\mathrm{T}} \right\}^{\mathrm{T}}
\tag{24}
$$

and enforcing it to be a true solution, this vector should ideally be equivalent to $\mathbf{0}$. This enables the determination of the $\boldsymbol{\beta}_j$ coefficients through various optimization techniques, such as least−square (LS) for linear problems [48] and iterative least−square (ILS) for non−linear problems [49].

If the iterative least−square method is necessary, the update of the estimations for the unknowns occurs at each iteration as follows:

$$
\boldsymbol{\beta}_{k+1} = \boldsymbol{\beta}_k + \Delta\boldsymbol{\beta}_k
\tag{25}
$$

where $\boldsymbol{\beta}$ represents the augmented vector containing all the vectors $\boldsymbol{\beta}_j$ (and potentially the square root of the mapping coefficient $b$, if the final time is unknown), with the subscript $k$ denoting the current iteration. In general, the term $\Delta\boldsymbol{\beta}_k$ can be determined by executing the conventional linear least−square method at each iteration of the iterative least−square procedure:

$$
\Delta\boldsymbol{\beta}_k = - \left( \mathbb{J}(\boldsymbol{\beta}_k)^{\mathrm{T}} \mathbb{J}(\boldsymbol{\beta}_k) \right)^{-1} \mathbb{J}(\boldsymbol{\beta}_k)^{\mathrm{T}} \mathbb{L}(\boldsymbol{\beta}_k)
\tag{26}
$$

where $\mathbb{J}$ represents the Jacobian matrix, encompassing the derivatives of the losses concerning all the unknowns. One may opt to compute the Jacobian manually or utilize computational tools like Symbolic or Automatic Differentiation routines. The iterative process persists until the following condition is satisfied:

$$
L_2[\mathbb{L}(\boldsymbol{\beta}_k)] < \epsilon
\tag{27}
$$

where $\epsilon$ denotes a user−defined tolerance, and $L_2$ signifies the $L_2$ norm.

Once we are sure that the convergence is achieved by meeting the criterion $L_2[\mathbb{L}(\boldsymbol{\beta}_k)] < \epsilon$, we can continue to solve the non−linear ODE until the following criterion $L_2[\mathbb{L}(\boldsymbol{\beta}_{k+1})] > L_2[\mathbb{L}(\boldsymbol{\beta}_k)]$ is met. Doing so, the convergence is at least satisfied for the imposed prescribed tolerance and it is pushed to achieve the best accuracy until the round−off error appears. Consequently, the solution accuracy is maximized, allowing for the attainment of the best possible solution accuracy tailored to the specific ODE. Nevertheless, for highly non−linear problems, alternative algorithms can be leveraged to enhance the efficiency of the solution search. Examples include the Levenberg−Marquardt and the trust−region−reflective algorithms. Both of these algorithms are incorporated in the "lsqnonlin" function of MatLab, and they are indeed utilized in this study.

For the reader's convenience, Figure 1 provides a schematic illustrating how the proposed PoNN−based framework operates for solving generic OCPs. Summarizing, once the OCP is transformed into a TPBVP (step 1), each latent solution required for the TPBVP is approximated via the X−TFC CEs taking into account the ICs and BCs (steps 2, 3 and 4). At this point, also transversality conditions on the costates are eventually considered, since they represent ICs or BCs that are analytically satisfied via the CEs. Afterwards, the CEs

are substituted into the set of the ODEs building the unconstrained TPBVP (step 5), which are then written in their implicit forms to build the loss functions and the augmented loss vector (step 6). Algorithms to minimize the loss vector, such as LS or ILS, are then employed to update the output coefficients $\beta$ and train X−TFC (step 7) to learn the optimal solution (step 8).

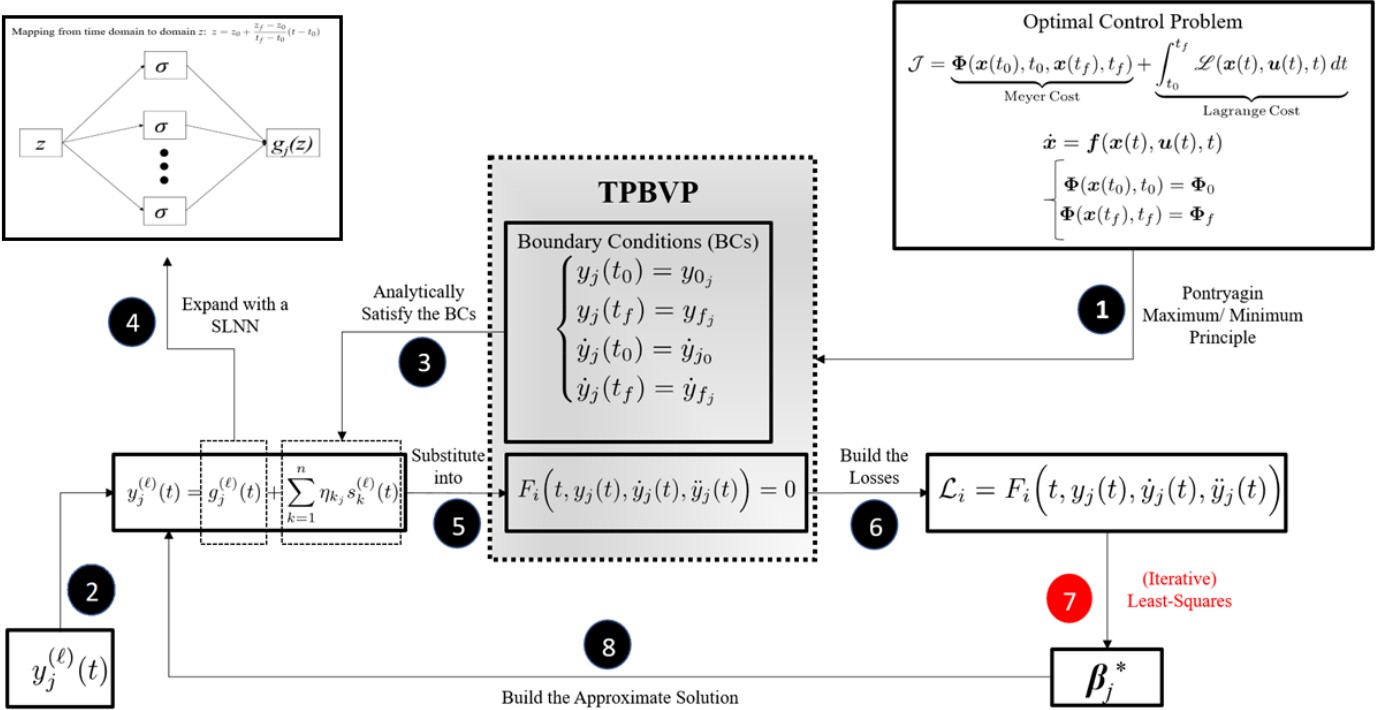

**Figure 1.** Schematic summarizing how PoNNs work for learning the solution of generic OCPs. The red font in step 7 indicates the optimization procedure to retrieve the optimal weights $\beta_j^*$.

## 3. Fuel−Optimal Control Problems

Typically, in a fuel−optimal control problem, the objective is to either maximize the final mass or minimize the fuel consumption. In this study, the latter approach is pursued. Hence, the cost function to be minimized is expressed as shown below [17]:

$$\min \mathcal{J} = \frac{1}{I_{sp}g_0} \int_{t_0}^{t_f} T dt \tag{28}$$

where $T$ and $I_{sp}$ denote the thrust magnitude supplied by the spacecraft engine and its specific impulse, respectively. Here, $g_0 = 9.80665$ m/s$^2$, and $t_0$ and $t_f$ represent the initial and final times (in this study, $t_0 = 0$). Traditionally, in such problems, the thrust vector is expressed as $\mathbf{T} = T\hat{\boldsymbol{\alpha}} = T_{max}\delta\hat{\boldsymbol{\alpha}}$, where $T_{max}$ denotes the maximum available thrust, $\delta \in [0, 1]$ signifies the throttle factor (also referred to as the engine thrust ratio), and $\hat{\boldsymbol{\alpha}}$ (hereafter simply denoted as $\boldsymbol{\alpha}$) represents the direction of the thrust (unit vector). Consequently, Equation (28) is transformed into the following:

$$\min \mathcal{J} = \frac{T_{max}}{I_{sp}g_0} \int_{t_0}^{t_f} \delta dt \tag{29}$$

In this study, the final time $t_f$ is presumed to be fixed and determined by the user. For rendezvous and landing trajectory scenarios, the boundary constraints are defined by fixed initial and final states, thus

$$\boldsymbol{r}(t_0) = \boldsymbol{r}_0; \quad \boldsymbol{v}(t_0) = \boldsymbol{v}_0; \quad m(t_0) = m_0 \tag{30}$$

$$\boldsymbol{r}(t_f) = \boldsymbol{r}_f; \quad \boldsymbol{v}(t_f) = \boldsymbol{v}_f \tag{31}$$

The following subsections will derive the TPBVPs associated to interplanetary and landing trajectories, respectively.

### 3.1. Interplanetary Trajectories

In the scenario of an interplanetary trajectory, where only the gravitational attraction of the Sun is considered, the equations of motion for a spacecraft, expressed in the heliocentric frame, are provided as follows [17]:

$$\dot{\mathbf{r}} = \mathbf{v} \tag{32}$$

$$\dot{\mathbf{v}} = -\frac{\mu}{r^3}\mathbf{r} + \frac{T_{max}\delta}{m}\boldsymbol{\alpha} \tag{33}$$

$$\dot{m} = -\frac{T_{max}\delta}{I_{sp}g_0} \tag{34}$$

where $\mu = 1.32712440018 \times 10^{11}$ km$^3$/s$^2$ represents the Sun's gravitational constant, $\mathbf{r}$ and $\mathbf{v}$ stand for the spacecraft's position and velocity vectors, $r$ denotes the norm of the position vector, and $m$ indicates the mass of the spacecraft. Using Equations (32)−(34), the Hamiltonian of the system can be written as

$$\mathcal{H} = \frac{T_{max}\delta}{I_{sp}g_0} + \boldsymbol{\lambda}_r^{\mathsf{T}}\boldsymbol{v} + \boldsymbol{\lambda}_v^{\mathsf{T}}\left[-\frac{\mu}{r^3}\mathbf{r} + \frac{T_{max}\delta}{m}\boldsymbol{\alpha}\right] - \lambda_m\frac{T_{max}\delta}{I_{sp}g_0} =$$
$$= -\frac{T_{max}}{I_{sp}g_0}\left(-\frac{I_{sp}g_0\boldsymbol{\lambda}_v^{\mathsf{T}}\boldsymbol{\alpha}}{m} + \lambda_m - 1\right)\delta + \boldsymbol{\lambda}_r^{\mathsf{T}}\boldsymbol{v} + \boldsymbol{\lambda}_v^{\mathsf{T}}\left(-\frac{\mu}{r^3}\mathbf{r}\right) \tag{35}$$

where $\boldsymbol{\lambda}_r$, $\boldsymbol{\lambda}_v$ and $\lambda_m$ represent the costates of position, velocity, and mass, respectively. By applying Pontryagin's Minimum Principle (PMP), the optimal thrust direction and magnitude are

$$\boldsymbol{\alpha}^* = -\frac{\boldsymbol{\lambda}_v}{||\boldsymbol{\lambda}_v||} \tag{36}$$

$$\delta^*(S) = \begin{cases} \delta = 0 & \text{if } S < 0 \\ \delta = 1 & \text{if } S > 0 \\ \delta \in [0,1] & \text{if } S = 0 \end{cases} \tag{37}$$

where $S$ is the switching function associated to the fuel−optimal problem, and it is defined as the term in parenthesis in Equation (35) that multiplies the control $\delta$:

$$S = \frac{I_{sp}g_0||\boldsymbol{\lambda}_v||}{m} + \lambda_m - 1 \tag{38}$$

Please note that Equation (38) has been obtained by substituting the optimal thrust direction $\boldsymbol{\alpha}^*$ of Equation (36) into Equation (35). The third case in Equation (37) seldom occurs, since $S$ is zero only at finite isolated points [17]. The additional first−order necessary conditions for optimality related to the costates are

$$\dot{\boldsymbol{\lambda}}_r = \frac{\mu}{r^3}\boldsymbol{\lambda}_v - \frac{3\mu\,\mathbf{r}\cdot\boldsymbol{\lambda}_v}{r^5}\mathbf{r} \tag{39}$$

$$\dot{\boldsymbol{\lambda}}_v = -\boldsymbol{\lambda}_r \tag{40}$$

$$\dot{\lambda}_m = -\frac{T_{max}\delta}{m^2}||\boldsymbol{\lambda}_v|| \tag{41}$$

Since the final mass is free, the transversality condition on the mass costate has to be imposed. Thus,

$$\lambda_m(t_f) = 0 \tag{42}$$

Finally, Equations (32)−(34) and (39)−(42), together with the boundary conditions expressed by Equations (30) and (31), represent the set of equations that build the TPBVP associated to the fuel−optimal control problem. Traditionally, this TPBVP can be solved via a shooting method which provides the initial costates $[\lambda_r(t_0); \lambda_v(t_0); \lambda_m(t_0)]$ that satisfy the equality constraints given by the following terminal conditions:

$$\mathbf{\Psi} = [\mathbf{r}(t_f) - \mathbf{r}_f, \mathbf{v}(t_f) - \mathbf{v}_f, \lambda_m(t_f)]^\top = \mathbf{0} \tag{43}$$

However, initiating the shooting technique necessitates an initial estimation for the initial costates. This requirement poses a drawback to the shooting method, as the solution to many problems is highly dependent on the initial guess. Moreover, challenges in solving fueloptimal problems emerge due to the discontinuities present in the optimal throttle function, as outlined in Equations (33), (34) and (41). Numerous numerical methods have been proposed to address this discontinuity and enhance solution accuracy. Among these methods, the smoothing technique is particularly appealing, involving the approximation of the discontinuous function $\delta^*(S)$ with a smooth function. In this study, the hyperbolic tangent function is adopted [31,32]. Consequently, if only the first two conditions of Equation (37) are considered, the following equation can be formulated [31,32]:

$$\delta^*(S) = \frac{1}{2}[1 + \text{sign}(S)] \approx \delta^*(S, \rho) = \frac{1}{2}\left[1 + \tanh\left(\frac{S}{\rho}\right)\right] \tag{44}$$

where $\rho$ denotes the smoothing level, which is also known as the smoothing parameter. It is notable that the approximation given by Equation (44) remains valid as $\rho \to 0$. This is why a common practice involves implementing a continuation procedure on the parameter $\rho$, gradually reducing its value toward 0 while maintaining solution accuracy. For $\rho$ values nearing 1, the hyperbolic tangent function offers a smooth approximation of $\delta^*(S)$. However, as $\rho$ diminishes, the slope of the hyperbolic tangent increases, approximating the discontinuous behavior of the sign function. It is important to note that the smoothing approximation of Equation (44) is specifically applicable to control bounded within the range $[0, 1]$. For the general bounded control input, the hyperbolic tangent approximation is represented as

$$\delta^*(S, \rho) = \frac{1}{2}\left[(\delta_l + \delta_u) + (\delta_u - \delta_l)\tanh\left(\frac{S - S_c}{\rho}\right)\right] \tag{45}$$

where $\delta_l$ and $\delta_u$ denote the lower and upper bounds of the control input, respectively, while $S_c$ signifies the switching point. It is noteworthy that in all the problems addressed in this study, the switching points consistently correspond to instances when the switching function crosses zero, indicating $S_c = 0$. Introducing the general approximation of Equation (45) allows to introduce also a minimum value of the control which is different from zero. Moreover, it is easy to see that for $\delta_u = 1$ and $\delta_l = 0$, Equation (45) provides Equation (44).

### 3.2. Landing Trajectories

In the context of landing trajectories, the dynamics equations delineated by Equations (32)−(34) remain applicable with an exception concerning the first term on the right−hand side of Equation (33). Typically, this term is replaced by a constant for landing on large planetary bodies, where $\mathbf{g}(\mathbf{r}) = \mathbf{g} = [0, 0, -g]$, with $g$ representing the constant gravitational acceleration along the vertical axis. However, for the more intricate scenario of landing on small planetary bodies, like asteroids and comets, where the gravitational field is highly irregular and the rotation period cannot be disregarded, this term is substituted with the following:

$$a_g = \mathbf{g}(\mathbf{r}) - 2\boldsymbol{\omega} \times \boldsymbol{v} - \boldsymbol{\omega} \times (\boldsymbol{\omega} \times \mathbf{r}) = \mathbf{g}(\mathbf{r}) + M\mathbf{v} + N\mathbf{r} \tag{46}$$

where $M$ and $N$ are defined as

$$M := \begin{bmatrix} 0 & 2\omega & 0 \\ -2\omega & 0 & 0 \\ 0 & 0 & 0 \end{bmatrix} \quad ; \quad N := \begin{bmatrix} \omega^2 & 0 & 0 \\ 0 & \omega^2 & 0 \\ 0 & 0 & 0 \end{bmatrix}, \tag{47}$$

One can note that the Coriolis and centrifugal accelerations have been added to take into account the rotation of the body having an angular velocity $\omega$. This means that in the case of landing on small planetary bodies, the equations of motion are considered to be written in a rotating reference frame, which is usually fixed with the body. There are many different ways to express the term $\mathbf{g}(\mathbf{r})$, such as via spherical harmonics, the mass dipole model [50], and the polyhedron model [51]. Another option that can simplify the problem, by removing the dependency on $\mathbf{r}$, is to consider a linearization of the gravitational acceleration as follows:

$$\mathbf{g}(\mathbf{r}) = \mathbf{g}(\mathbf{r}_0) + G(\mathbf{r}_0) \cdot (\mathbf{r} - \mathbf{r}_0) \tag{48}$$

where $\mathbf{r}_0$ is the initial position, $\mathbf{g}(\mathbf{r}_0) = \mathbf{g}_0$ and $G(\mathbf{r}_0) = G$ are the gravitational acceleration and the gravity gradient matrix computed at the initial time instant. Thus, the generic Hamiltonian of the problem becomes

$$\mathcal{H} = \frac{T_{max}\delta}{I_{sp}g_0} + \lambda_r^\mathsf{T} v + \lambda_v^\mathsf{T} \left[ \mathbf{g}(\mathbf{r}) + \frac{T_{max}\delta}{m}\alpha \right] - \lambda_m \frac{T_{max}\delta}{I_{sp}g_0} \tag{49}$$

If the linearized gravity is employed, the mass costate equation remains the same as Equation (41), whereas the position and velocity costates equations become

$$\dot{\lambda}_r = -(N + G)^\mathsf{T}\lambda_v \tag{50}$$

$$\dot{\lambda}_v = -\lambda_r - M^\mathsf{T}\lambda_v \tag{51}$$

In this work, where a constant gravitational acceleration is considered (i.e., when large planetary bodies are taken into account for the landing), the position and velocity costates equations become

$$\dot{\lambda}_r = \mathbf{0} \tag{52}$$

$$\dot{\lambda}_v = -\lambda_r \tag{53}$$

Generally, the TPBVP to solve for the landing is here represented by Equations (32) and (33) with the modification of Equations (34), (41), (48), (50) and (51), together with the boundary conditions expressed by Equations (30) and (31) and the transversality condition of Equation (42).

### 3.3. X−TFC Approximation of States/Costates

For both the interplanetary trajectories and landing problems, the PINN X−TFC is used within the PoNNs framework to learn the solution of the TPBVP. Therefore, the states and costates are approximated by using the constrained expressions, which are here derived for the boundary conditions $r(t_0) = r_0, v(t_0) = v_0, m(t_0) = m_0, r(t_f) = r_f$, and $v(t_f) = v_f$ as follows

$$r_i = \left(\sigma - \Omega_1 \sigma_0 - \Omega_2 \sigma_f - \Omega_3 \sigma_0' - \Omega_4 \sigma_f'\right)^{\mathrm{T}} \beta_i + \Omega_1 r_{0_i} + \Omega_2 r_{f_i} + \frac{\Omega_3 v_{0_i} + \Omega_4 v_{f_i}}{b^2} \tag{54}$$

$$v_i = b^2 \left[\left(\sigma' - \Omega_1' \sigma_0 - \Omega_2' \sigma_f - \Omega_3' \sigma_0' - \Omega_4' \sigma_f'\right)^{\mathrm{T}} \beta_i + \Omega_1' r_{0_i} + \Omega_2' r_{f_i} + \frac{\Omega_3' v_{0_i} + \Omega_4' v_{f_i}}{b^2}\right] \tag{55}$$

$$a_i = b^4 \left[\left(\sigma'' - \Omega_1'' \sigma_0 - \Omega_2'' \sigma_f - \Omega_3'' \sigma_0' - \Omega_4'' \sigma_f'\right)^{\mathrm{T}} \beta_i + \Omega_1'' r_{0_i} + \Omega_2'' r_{f_i} + \frac{\Omega_3'' v_{0_i} + \Omega_4'' v_{f_i}}{b^2}\right] \tag{56}$$

$$m = \left(\sigma - \Omega_1 \sigma_0\right)^{\mathrm{T}} \beta_m + \Omega_1 m_0 \tag{57}$$

$$\dot{m} = b^2 \left[\left(\sigma' - \Omega_1' \sigma_0\right)^{\mathrm{T}} \beta_m + \Omega_1' m_0\right] \tag{58}$$

where $i = 1, 2, 3$ and we define the notation $r_{f_i} = r_i(t_f)$ and $v_{f_i} = v_i(t_f)$ as the position and velocity components evaluated at the final time $t_f$, while the $\Omega_k(z)$ terms are called switching functions and are defined in Appendix A, and $b$ is defined according to the same procedure previously explained (the reader has to consider that the CEs switching functions, $\Omega_k(z)$, are different from the switching function $S$ defined before within the fuel−optimal problem). The constrained expressions for the costates are

$$\lambda_{r_i} = \sigma_r^{\mathrm{T}} \beta_{ri} \tag{59}$$

$$\dot{\lambda}_{r_i} = b^2 \sigma_r'^{\mathrm{T}} \beta_{ri} \tag{60}$$

$$\lambda_{v_i} = \sigma_v^{\mathrm{T}} \beta_{vi} \tag{61}$$

$$\dot{\lambda}_{v_i} = b^2 \sigma_v'^{\mathrm{T}} \beta_{v_i} \tag{62}$$

$$\ddot{\lambda}_{v_i} = b^4 \sigma_v''^{\mathrm{T}} \beta_{v_i} \tag{63}$$

$$\lambda_m = \left(\sigma - \Omega_1 \sigma_f\right)^{\mathrm{T}} \beta_{\lambda_m} + \Omega_1 \lambda_{m_f} \tag{64}$$

$$\dot{\lambda}_m = b^2 \left[\left(\sigma' - \Omega_1' \sigma_f\right)^{\mathrm{T}} \beta_{\lambda_m} + \Omega_1' \lambda_{m_f}\right] \tag{65}$$

It is important to notice that the expressions of the CEs switching functions $\Omega_k(z)$ are not always the same for all the constrained expressions above, but they change according to the constraints imposed on the corresponding variable. Indeed, for $r_i$, $v_i$ and $a_i$, the $\Omega_k(z)$ reported in Table A6 of Appendix A must be used. For $m$, $\dot{m}$, $\lambda_m$ and $\dot{\lambda}_m$, the $\Omega_k(z)$ reported in Table A1 of Appendix A must be used. One can note that the equality constraints appearing in the shooting function of Equation (43) are already satisfied a priori by the constrained expressions, thus simplifying the problem to be solved. Once the constrained expressions have been built, the loss functions have to be retrieved. These are obtained simply by writing the set of differential equations describing the TPBVP in their implicit form. One should consider that the whole problem can be eased (for both interplanetary and landing trajectories), since Equation (32) is implicitly satisfied by the constrained expression approximating $r_i$ and $v_i$. In addition, the transversality condition on the mass costate represented by Equation (42) can also be removed, since the boundary condition is satisfied by the constrained expressions associated to $\lambda_m$. Finally, if the interplanetary trajectory problem is taken into account, Equation (40) can also be removed from the losses, since it is satisfied by the constrained expression of $\lambda_{v_i}$. In fact, from the constrained expression of $\lambda_{v_i}$, $\lambda_{r_i}$ can be directly computed from the opposite of its first time derivative ($\lambda_r = -\dot{\lambda}_v$) and Equation (39) can be written as a function of $-\ddot{\lambda}_v$. This also means that the constrained expressions for $\lambda_{r_i}$ and $\dot{\lambda}_{r_i}$ can actually be removed, reducing the number of unknowns of the problem. To summarize, the unknowns for each problem are shown below:

- Interplanetary trajectory: $\Xi = \{\beta_1 \quad \beta_2 \quad \beta_3 \quad \beta_m \quad \beta_{\lambda_{v1}} \quad \beta_{\lambda_{v2}} \quad \beta_{\lambda_{v3}} \quad \beta_{\lambda_m}\}^{\mathrm{T}}$;

- Landing: $\Xi = \left\{ \begin{array}{ccccccccccc} \boldsymbol{\beta}_1 & \boldsymbol{\beta}_2 & \boldsymbol{\beta}_3 & \boldsymbol{\beta}_m & \boldsymbol{\beta}_{\lambda_{r1}} & \boldsymbol{\beta}_{\lambda_{r2}} & \boldsymbol{\beta}_{\lambda_{r3}} & \boldsymbol{\beta}_{\lambda_{v1}} & \boldsymbol{\beta}_{\lambda_{v2}} & \boldsymbol{\beta}_{\lambda_{v3}} \\ \boldsymbol{\beta}_{\lambda_m} & & & & & & & & & \end{array} \right\}^{\mathrm{T}}.$

## 4. Results

The methodology elucidated in the preceding sections is employed herein to address two distinct problems: a fuel−optimal interplanetary rendezvous trajectory from Earth to Mars, employing low−thrust maneuvers, and a fuel−optimal landing on Mars. In particular, in the landing problem on a large planetary body under the assumption of a constant gravity (i.e., short times of flight), the number of control switches is known a priori and follows the structure $T_{max} - T_{min} - T_{max}$. This happens because the time derivative of the switching function $\dot{S}$ is proved to change sign at most once, leading to at most two changes of sign of $S$. Therefore, the control profile presents at most three subarcs with two switches. On the other hand, in case of a central gravity field assumption (e.g., the interplanetary transfer), $\dot{S}$ can change sign more than once, leading to possibly more than three subarcs in the control profile. For more information and detailed theoretical proofs, the reader can refer to Ref. [52]. For both the problems, the time of flight is considered fixed. The initial and final conditions of the two problems, together with the parameters associated with the spacecraft engine, are reported in Table 1. For the interplanetary problem, the same values used in Ref. [31] are employed, while for the landing, the same parameters of Ref. [46] are exploited. Furthermore, Table 1 also reports the values employed to make the problem dimensionless: $\bar{R}$ for distance, $\bar{t}$ for time, and $\bar{m}$ for mass. All the other quantities can be made dimensionless according to a combination of the three mentioned parameters. Finally, the X−TFC parameters are shown in Table 2. One can note that if Chebyshev Neural Networks are employed as activation functions, the input weights and bias ($w_q$ and $b_q$, respectively) are constant values set equal to one and zero, respectively. All the simulations have been implemented in Matlab R2023b and ran with an Intel Core i7−9700 CPU PC with 64 GB of RAM.

**Table 1.** Parameters employed for the simulations.

| Problem | Interplanetary Trajectory | Landing |
|---|---|---|
| $r_0$ | $[-140{,}699{,}693, -51{,}614{,}428, 980]$ km | $[-200, 100, 1500]$ m |
| $v_0$ | $[9.774596, -28.07828, 4.337725 \times 10^{-4}]$ km/s | $[85, -50, -65]$ m/s |
| $m_0$ | 1000 kg | 1905 kg |
| $r_f$ | $[-172{,}682{,}023, 176{,}959{,}469, 7{,}948{,}912]$ km | $[0, 0, 0]$ m |
| $v_f$ | $[-16.427384, -14.860506, 9.21486 \times 10^{-2}]$ km/s | $[0, 0, 0]$ m/s |
| $t_f$ | 348.795 days | 44.823 s |
| $T_{max}$ | 0.5 N | 13,258.18 N |
| $T_{min}$ | 0 N | 4971.81 N |
| $I_{sp}$ | 2000 s | 225 s |
| $\bar{R}$ | $149.59787 \times 10^6$ km | 1516.57 m |
| $\bar{t}$ | 58.132 days | 44.823 s |
| $\bar{m}$ | 1000 kg | 1905 kg |

**Table 2.** X−TFC parameters used within PoNNs.

| Problem | $n$ | $L$ | Activation Function ($\sigma$) | Range for $w_q$ | Range for $b_q$ |
|---|---|---|---|---|---|
| Interplanetary | 140 | 70 | Chebyshev Polynomials | 1 (constant) | 0 (constant) |
| Landing | 120 | 30 | Chebyshev Polynomials | 1 (constant) | 0 (constant) |

### 4.1. Interplanetary Trajectory

In tackling this problem, the smoothing parameter $\rho$ has been progressively decreased from 1 to $10^{-10}$ over 20 iterations, employing a logarithmically spaced vector. Regarding the

initial guesses for the unknown coefficients of the constrained expressions, they have been randomly set from a standard uniform distribution $\mathcal{U}(0,1)$ for $[\boldsymbol{\beta}_1, \boldsymbol{\beta}_2, \boldsymbol{\beta}_3, \boldsymbol{\beta}_{\lambda_{v1}}, \boldsymbol{\beta}_{\lambda_{v2}}, \boldsymbol{\beta}_{\lambda_{v3}}]$. For the mass and mass costate, $\boldsymbol{\beta}_m$ and $\boldsymbol{\beta}_{\lambda_m}$ have been computed so that the first guess solutions are a constant value equal to $m_0$ for $m$ and equal to one for $\lambda_m$. The algorithm employed for the training of PoNNs is the Levenberg−Marquardt. The results regarding the switching function $S$ and the throttle function are shown in Figure 2 together with a comparison carried out via the shooting method. As can be seen, the PoNNs are able to compute the time location of the switching points well enough. However, the solution for the control stills appears different from the one obtained via the shooting technique, especially in the discontinuities of the control. In order to increase the accuracy, a last step is carried out to better refine the solution and eventually increase the accuracy. Therefore, once the switch intervals are identified, 100 discretization points per switch interval are added to $n$, and the number of neurons $L$ has been increased to 80. Hence, the X−TFC framework has been run again, and the results after the refinement of the solution are shown in Figures 3−5. In particular, the interplanetary transfer and the direction of the thrust are illustrated in Figures 3 and 4, respectively. The switching function $S$, the control, the mass and mass costate are shown in Figure 5 together with the comparison with the shooting method. The new refined solution is now very close to the one obtained via the shooting method despite a little difference in the last switch of the control thrust. In particular, the fuel consumption computed via PoNNs is 396.85 kg, which differs by 0.79 kg from the minimum solution obtained via the shooting method (396.06 kg). This value is also compliant with the one reported in Ref. [31]. The results obtained with the current simulation are very promising, since the PINN is able to learn the solution together with the correct number of control switches and the switching times without any previous knowledge. Moreover, this is carried out without splitting the time domain into multiple segments as it was shown for a fuel−optimal landing on Mars solved via TFC in a previous work [46].

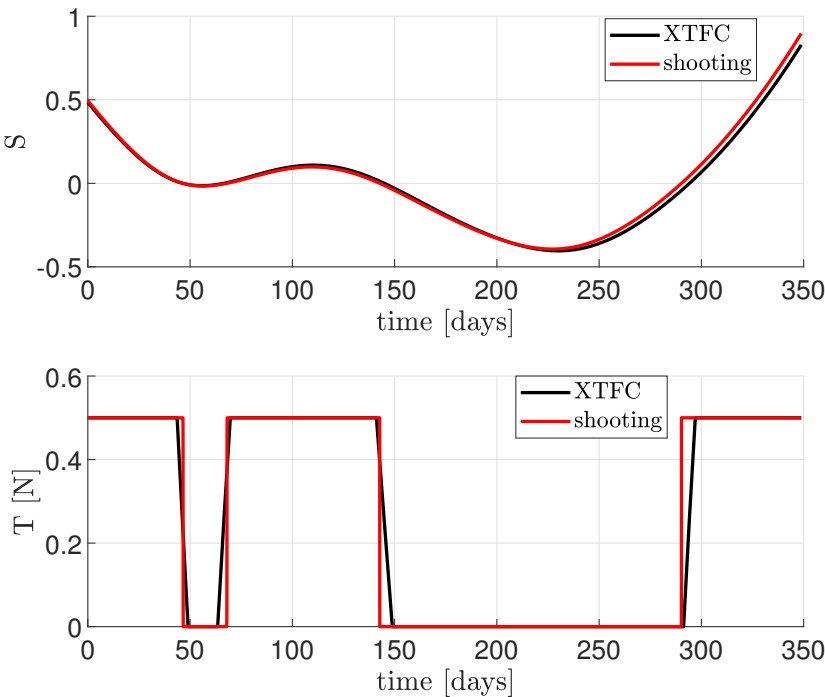

**Figure 2.** Solution for the Earth−Mars interplanetary transfer without solution refinement.

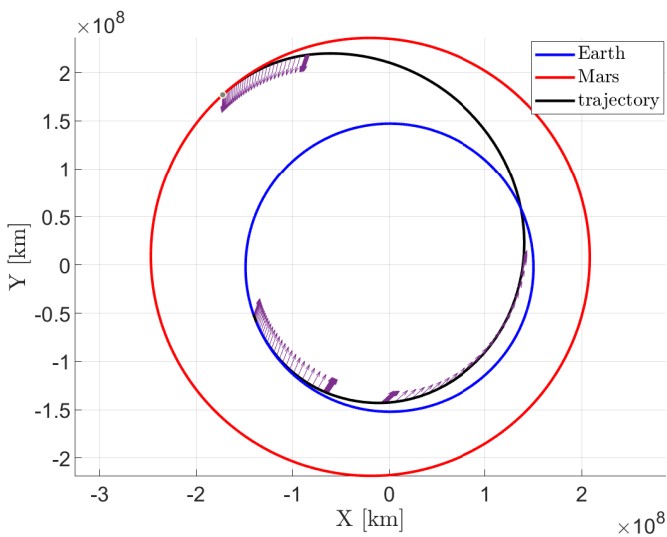

**Figure 3.** Earth−Mars interplanetary transfer trajectory after solution refinement. The purple arrows indicates the direction of the thrust.

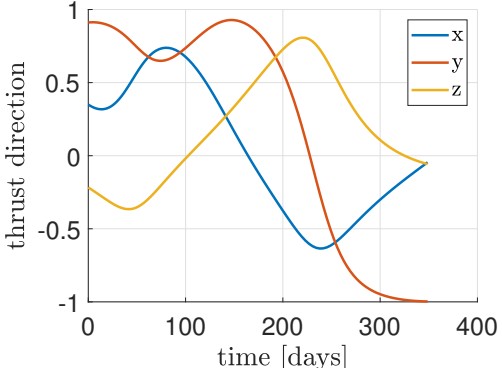

**Figure 4.** Thrust direction for the Earth−Mars interplanetary transfer after solution refinement.

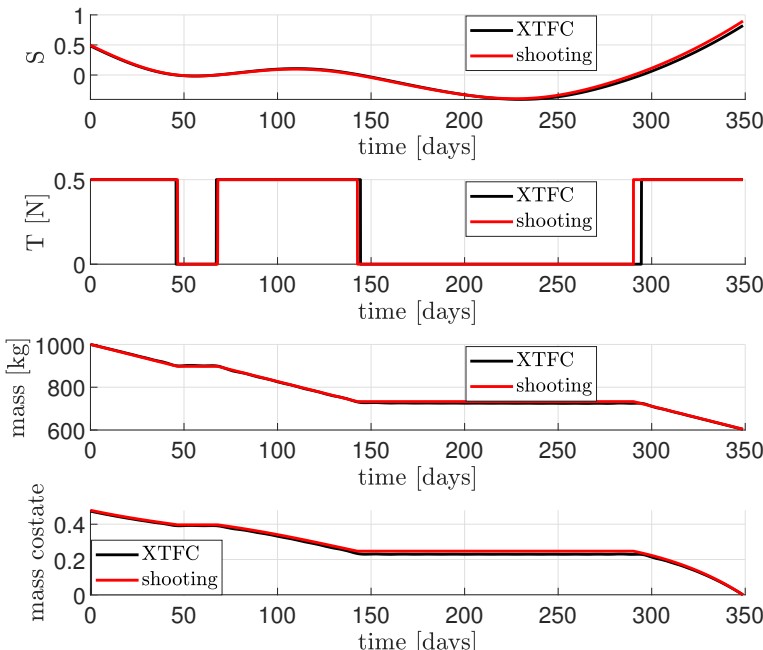

**Figure 5.** Solution for the Earth−Mars interplanetary transfer after solution refinement.

### 4.2. Landing Trajectory

For this problem, the smoothing parameter $\rho$ has been decreased from 1 to $10^{-10}$ with 20 iterations using a logarithmically spaced vector. For what concerns the initial guesses of the unknown coefficients of the constrained expressions, they have been set randomly for $[\beta_1, \beta_2, \beta_3, \beta_{\lambda_{r1}}, \beta_{\lambda_{r2}}, \beta_{\lambda_{r3}}, \beta_{\lambda_{v1}}, \beta_{\lambda_{v2}}, \beta_{\lambda_{v3}}]$. For the mass and mass costate, $\beta_m$ and $\beta_{\lambda_m}$ have been computed so that the first$-$guess solutions are a constant value equal to $m_0$ for $m$ and equal to 1 for $\lambda_m$. The algorithm employed is the trust$-$region$-$reflective algorithm. The result regarding the time history of the control thrust is shown in Figure 6 together with a comparison carried out via the adaptive Gaussian quadrature collocation method, as implemented in GPOPS$-$II [53]. Even in this case, PoNNs are able to compute the time location of the switching points well enough. In order to increase the accuracy, the refinement of the solution is carried out. Therefore, once the switch intervals are identified, 100 discretization points per switch interval are added to $n$, whereas the number of neurons $L$ has not been increased. Hence, the PoNNs framework has been run again, and the results after the refinement of the solution are shown in Figures 7$-$9. In particular, the landing trajectory and the direction of the thrust are illustrated in Figures 7 and 8, respectively. The switching function $S$, the control, the mass and mass costate are shown in Figure 9, together with the comparison with GPOPS$-$II. The new refined solution is now very close to the one obtained via GPOPS$-$II, despite there being a little difference in the switching points. In particular, the fuel consumption computed via PoNNs is 268.30 kg, which differs by 0.37 kg from the minimum solution obtained via GPOPS$-$II (267.93 kg). This value is also similar to the one reported in Ref. [46], where the fuel consumption is shown to be 275.205 kg. Nevertheless, the approach pursued in this work avoids the a priori knowledge of the number of switches and also the necessity to split the domain to handle discontinuities in the control, as shown in [46]. Another advantage of the proposed approach with respect to Ref. [46] is that an additional optimizer to obtain the switching times, represented by an outer loop on top of the TFC solver, is not required anymore, since the entire solution is computed with the only PoNN framework. However, the method based on the split domain is useful to increase the accuracy of the loss functions in correspondence of the switches. In fact, with the proposed approach, the loss functions present some jumps when the discontinuities are present. This indicates that the two methods can actually be used in combination: the current one to detect the first guess solution with the number of switches and their time location, and the one based on the domain splitting to improve accuracy and performances.

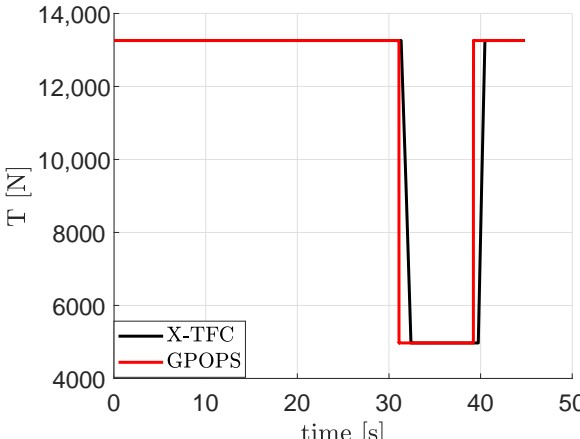

**Figure 6.** Solution for the the Mars landing before the solution refinement.

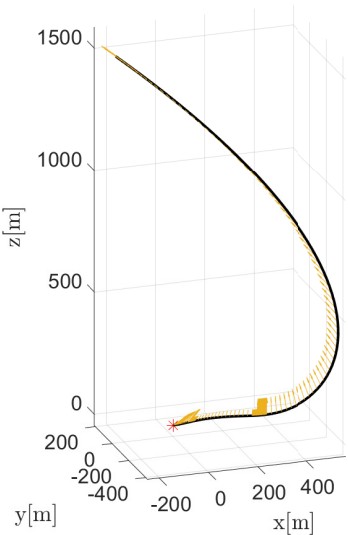

**Figure 7.** Mars landing trajectory after solution refinement.

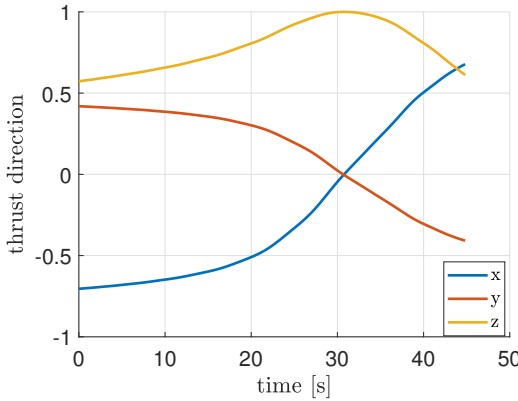

**Figure 8.** Thrust direction for the Mars landing after solution refinement.

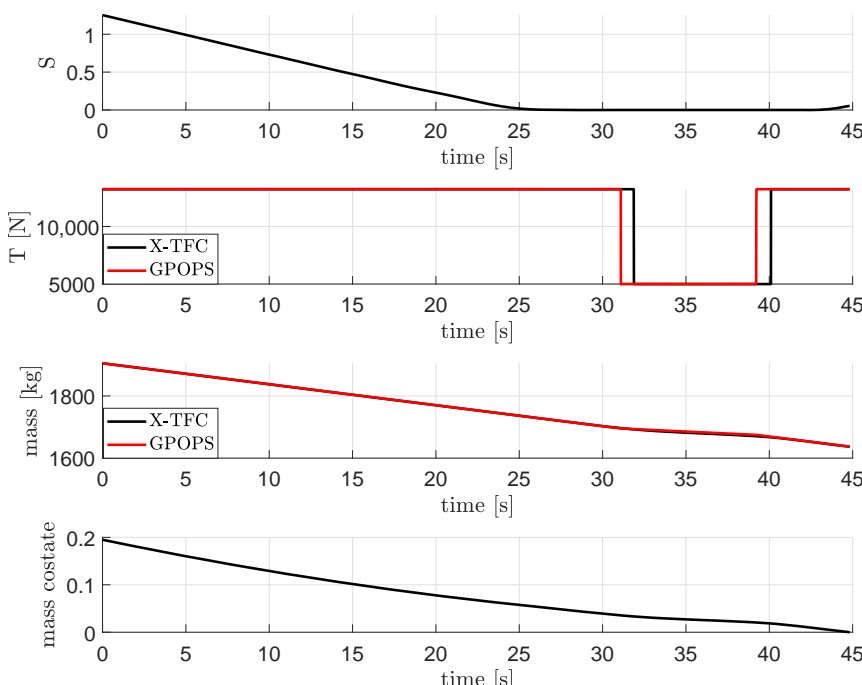

**Figure 9.** Solution for the Mars landing after solution refinement.

## 5. Discussion

The following advantages can be highlighted when using PoNNs to solve OCPs similar to the ones proposed in this paper:

- In the examined scenarios, the framework demonstrates convergence to the solution even with random initialization of the unknowns. This capability stands as a notable advantage, considering the well−known impact of initial guesses on the results of optimization when employing traditional deterministic algorithms.

- Following the learning process of the optimal control on the training points, an analytical representation via CEs is derived. Consequently, this representation enables the evaluation of the control at points unseen during the training, such as test points, without necessitating interpolation techniques or additional computational efforts.

- Another notable advantage of the proposed framework lies in its potential for future extensions. Specifically, the network training can be conducted in a data−physics−driven manner. In this study, the PoNNs are utilized solely to learn the solution of the ODE system, employing a physics−driven training approach where the loss function solely considers the residual of the ODEs. Consequently, no labeled data from a dataset are utilized during the training process. However, in future endeavors, data can be integrated into the training process by incorporating an additional term in the loss function, which relates to the mean−square error between the NN predictions and the labeled data. This distinctive feature holds the potential to enhance the robustness of the proposed novel approach in solving challenging OCPs. Moreover, optimal trajectories generated using alternative techniques for solving OCPs can be sampled to generate data, which can then be included as additional terms within the PoNNs loss function. This approach can effectively bolster and streamline the learning process with data. Furthermore, this methodology can be extended to incorporate real−world data, which inherently contains information about perturbations and/or unmodeled terms of the actual dynamics. These elements may not be fully captured in the simulated dynamics modeled via differential equations.

## 6. Conclusions

In this work, Pontryagin Neural Networks (PoNNs) are utilized to learn the optimal control in fuel−optimal problems with a focus on employing the innovative Extreme Theory of Functional Connections (X−TFC) within the PINN framework. One key advantage of employing X−TFC lies in the a priori satisfaction of boundary constraints through constrained expressions, which is a feature inherited from the original Theory of Functional Connections (TFC). Furthermore, the methodology introduced in this work incorporates the hyperbolic tangent function to approximate the sign function emerging from the optimality conditions of the PMP concerning the thrust magnitude. This strategic utilization enhances the framework's capability to handle discontinuities in the optimal control, contributing to its efficacy in solving complex optimization problems. A continuation procedure on the smoothing parameter, appearing in the hyperbolic tangent function, is carried out to slowly and accurately approach the discontinuous solution for the control. This smoothing technique allows for accurately learning the states−costates solution of the associated TPBVP. The proposed approach has been applied to a fixed time fuel−optimal trajectory from Earth to Mars with low−thrust propulsion, where the number of switches is not known a priori, and a landing trajectory on Mars. The results show good accuracy and the feasibility of PoNN to catch discontinuities in those problems where the control appears linearly in the Hamiltonian, thus leading to a discontinuous control. Moreover, the proposed approach has been compared with state−of−the−art techniques to solve optimal control problems, such as the shooting method and adaptive Gaussian quadrature collocation technique (as implemented in GPOPS−II), achieving comparable results. With respect to the past works related to the original TFC [46], the fuel−optimal problem is here solved without splitting the time domain in multiple segments according to the switching points. In fact, even though with this last approach, more accurate performances can be

obtained in terms of the precision of the dynamics and the optimality of the results, one should known a priori the number of switches, which is usually unknown for most of the problems.

For future research, in order to achieve better and more accurate results, the proposed method can be used in combination with the approach employing the split domain. Indeed, the current methodology can be used to first discover the number of switches and where they are located in the time domain, and then the split domain technique can be carried out to refine the solution and improve its accuracy. Furthermore, more complex OCPs, involving several control switches and multiple revolutions, should be considered. As seen in this paper and other works in the literature, it is worth carrying out further investigations about solving OCPs via NNs (in this case PoNNs), because they seem to be effective and helpful also for the real−time onboard generation of optimal trajectories.

**Author Contributions:** Conceptualization: A.D. and R.F.; methodology: A.D.; software: A.D.; validation: A.D.; formal analysis: A.D.; investigation: A.D.; resources: A.D. and R.F.; writing—original draft preparation: A.D.; writing—review and editing: A.D. and R.F.; visualization: A.D.; supervision: R.F. All authors have read and agreed to the published version of the manuscript.

**Funding:** This research received no external funding.

**Data Availability Statement:** Data sharing is not applicable to this article.

**Acknowledgments:** The authors would like to acknowledge Kristopher Drozd for providing useful advice regarding the code implementation of the X-TFC algorithm.

**Conflicts of Interest:** The authors declare no conflicts of interest.

## Appendix A. Switching Functions

The detailed derivation of the switching functions can be found in Ref. [46]. Here, for clarity, we provide the switching functions for different cases. By defining $\Delta z = z_f - z_0$ and $z_* = z - z_0$, the switching functions for a constrained expression with one constraint on $f$ are provided in Table A1. The switching functions for a constrained expression with one constraint on $f'$ are given in Table A2. The switching functions for a constrained expression with two constraints on $f$ are presented in Table A3. The switching functions for a constrained expression with two constraints, one on $f$ and one on $f'$, are given in Table A4. The switching functions for a three−constraints−constrained expression, with two constraints on $f$ and one on $f'$, are provided in Table A5. The switching functions for a four−constraints−constrained expression, with two constraints on $f$ and two constraints on $f'$, are outlined in Table A6.

**Table A1.** Switching functions for a constrained expression with one constraint, with the constraint of $f$, defined on the domain of $z \in [z_0, z_f]$.

|  | Initial/Final Value $\Omega_1(z)$ |
| --- | --- |
| $(\cdot)$ | 1 |
| $\dfrac{d}{dz}(\cdot)$ | 0 |

**Table A2.** Switching functions for a constrained expression with one constraint, with the constraint of $f'$, defined on the domain of $z \in [z_0, z_f]$.

|  | Initial/Final Value $\Omega_1(z)$ |
| --- | --- |
| $(\cdot)$ | $z$ |
| $\dfrac{d}{dz}(\cdot)$ | 1 |

**Table A3.** Switching functions for a constrained expression with two constraints, with both constrains on $f$, defined on the domain of $z \in [z_0, z_f]$.

|  | Initial Value $\Omega_1(z)$ | Final Value $\Omega_2(z)$ |
|---|---|---|
| $(\cdot)$ | $\dfrac{z_f - z}{\Delta z}$ | $\dfrac{z - z_0}{\Delta z}$ |
| $\dfrac{d}{dz}(\cdot)$ | $-\dfrac{1}{\Delta z}$ | $\dfrac{1}{\Delta z}$ |
| $\dfrac{d^2}{dz^2}(\cdot)$ | $0$ | $0$ |

**Table A4.** Switching functions for a constrained expression with two constraints, with one constraint on $f$ and one on $f'$, defined on the domain of $z \in [z_0, z_f]$.

|  | Initial Value $\Omega_1(z)$ | Initial Derivative Value $\Omega_2(z)$ |
|---|---|---|
| $(\cdot)$ | $1$ | $z - z_0$ |
| $\dfrac{d}{dz}(\cdot)$ | $0$ | $1$ |
| $\dfrac{d^2}{dz^2}(\cdot)$ | $0$ | $0$ |

**Table A5.** Switching functions for a constrained expression with three constraints, with two constraints on $f$ and one on $f'$, defined on the domain of $z \in [z_0, z_f]$.

|  | Initial Value $\Omega_1(z)$ | Final Value $\Omega_2(z)$ | Initial Derivative $\Omega_3(z)$ |
|---|---|---|---|
| $(\cdot)$ | $\dfrac{(z_f - z)(z - 2z_0 + z_f)}{\Delta z^2}$ | $\dfrac{(z - z_0)^2}{\Delta z^2}$ | $\dfrac{(z - z_0)(z_f - z)}{\Delta z}$ |
| $\dfrac{d}{dz}(\cdot)$ | $\dfrac{-2(z - z_0)}{\Delta z^2}$ | $\dfrac{2(z - z_0)}{\Delta z^2}$ | $\dfrac{-2z + z_0 + z_f}{\Delta z}$ |
| $\dfrac{d^2}{dz^2}(\cdot)$ | $\dfrac{-2}{\Delta z^2}$ | $\dfrac{2}{\Delta z^2}$ | $\dfrac{-2}{\Delta z}$ |

**Table A6.** Switching functions for a constrained expression with four constraints, with two constraints on $f$ and two constraints on $f'$, defined on the domain of $z \in [z_0, z_f]$.

|  | Initial Value $\Omega_1(z)$ | Final Value $\Omega_2(z)$ | Initial Derivative $\Omega_3(z)$ | Final Derivative $\Omega_4(z)$ |
|---|---|---|---|---|
| $(\cdot)$ | $1 + \dfrac{2z_*^3}{\Delta z^3} - \dfrac{3z_*^2}{\Delta z^2}$ | $-\dfrac{2z_*^3}{\Delta z^3} + \dfrac{3z_*^2}{\Delta z^2}$ | $z_* + \dfrac{z_*^3}{\Delta z^2} - \dfrac{2z_*^2}{\Delta z}$ | $\dfrac{z_*^3}{\Delta z^2} - \dfrac{z_*^2}{\Delta z}$ |
| $\dfrac{d}{dz}(\cdot)$ | $\dfrac{6z_*^2}{\Delta z^3} - \dfrac{6z_*}{\Delta z^2}$ | $-\dfrac{6z_*^2}{\Delta z^3} + \dfrac{6z_*}{\Delta z^2}$ | $1 + \dfrac{3z_*^2}{\Delta z^2} - \dfrac{4z_*}{\Delta z}$ | $\dfrac{3z_*^2}{\Delta z^2} - \dfrac{2z_*}{\Delta z}$ |
| $\dfrac{d^2}{dz^2}(\cdot)$ | $\dfrac{12z_*}{\Delta z^3} - \dfrac{6}{\Delta z^2}$ | $-\dfrac{12z_*}{\Delta z^3} + \dfrac{6}{\Delta z^2}$ | $\dfrac{6z_*}{\Delta z^2} - \dfrac{4}{\Delta z}$ | $\dfrac{6z_*}{\Delta z^2} - \dfrac{2}{\Delta z}$ |

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
