# Peer review of "Learning Fuel-Optimal Trajectories for Space Applications via Pontryagin Neural Networks"

_aerospace, doi:10.3390/aerospace11030228_

Round 1

Reviewer 1 Report

Comments and Suggestions for Authors

In this paper, the authors propose a method for solving the Two Point Boundary Value Problem (TPBVP), based on a heuristic search for a solution using neural networks. The advantage of the proposed method is the possibility of solving the TPBVP:

  regardless of the initial approximation,

- with the unknown number of switches of the control and their temporal location.

These advantages are demonstrated on model problems by comparing TPBVP solutions obtained by the proposed method with solutions by methods that provide higher accuracy.

‘Model’ because, at least in the second problem considered, not all acting forces are taken into account: the influence of aerodynamic forces due to the presence of an atmosphere on Mars is not taken into account.

Therefore, it is more correct to talk about the successful application of the proposed method to determine the initial approximation for solving the problem in a more accurate formulation by other methods.

 There are a number of minor comments to the presented work:

P. 8, Ln 283: The symbol for prescribed tolerance differs from used in formula (27).

P. 15, LL 476-478: A comparison with the previous solution of the optimal landing problem should probably be given in section 4.2. Landing Trajectory.

P. 17, LL 494-495: Incorrect reference to figures 7, 8 that they describe interplanetary flight, while they refer to landing on Mars.

  The paper can be published after taking into account the comments.

Reviewer 2 Report

Comments and Suggestions for Authors

Dear authors,

Please find blow my comments.

The paper subject is related to the Optimal Control problem and provides a new indirect method to    solve these kind of problems. The theory behind this method relies on the Pontryagin principle, Extreme Theory of connections and Extreme Learning Machine, (ELM). The key idea is to approximate the free function gj(l) with a single layer neural network, trained with an ELM algorithm. The main advantages of this method are underlined in Section  5 of the article.

However, several minor modification are needed such that this paper to  be accepted for publishing.

Comments on the Quality of English Language

Some misprints were found or minor language improvements are needed.

Round 2

Reviewer 2 Report

Comments and Suggestions for Authors
